# Evaluation of the Antimicrobial Activity of an Extract of *Lactobacillus casei*-Infected *Hermetia illucens* Larvae Produced Using an Automatic Injection System

**DOI:** 10.3390/ani10112121

**Published:** 2020-11-16

**Authors:** Kyu-Shik Lee, Eun-Young Yun, Tae-Won Goo

**Affiliations:** 1Department of Pharmacology, College of Medicine, Dongguk University, Gyeongju 38766, Korea; there1@dongguk.ac.kr; 2Department of Integrative Bio-industrial Engineering, Sejong University, Seoul 05006, Korea; yuney@sejong.ac.kr; 3Department of Biochemistry, College of Medicine, Dongguk University, Gyeongju 38766, Korea

**Keywords:** antimicrobial peptide, *Hermetia illucens* larvae, preservatives, automatic mass injection system

## Abstract

**Simple Summary:**

In this investigation, an automatic mass-injection system was developed to produce an extract of *Lactobacillus casei*–infected *Hermetia illucens* larvae (HIL) at low cost. The extract produced was found to be a novel natural antibiotic candidate with a wide range of applications, especially in the food, animal feed, and medicinal industries.

**Abstract:**

In the present study, we developed an automatic mass-injection system (AMIS) to produce an extract of infected *H. illucens* larvae (iHIL-E) and then evaluated antimicrobial peptide (AMP) expressions and assessed the antimicrobial activity of iHIL-E against various pathogens and *Lactobacillus* species. AMP gene expressions were assessed by real-time quantitative polymerase chain reaction (PCR) and the antimicrobial activities of iHIL-E were estimated using a radial diffusion assay and by determining minimal inhibitory concentrations. Results showed that the antimicrobial activity of HIL extract was effectively enhanced by *L. casei* infection and that the gene expressions of cecropin 3 and defensin 3 (antimicrobial peptides) were up-regulated. iHIL-E also prevented the growths of *Enterococcus faecalis*, *Streptococcus mutans*, and *Candida vaginitis* (MICs 200, 500, and 1000 µg/100 µL, respectively) and demonstrated high protease resistance. Moreover, the growths of methicillin-resistant *Staphylococcus aureus*, antibiotic-resistant *Pseudomonas aeruginosa* and AMP-resistant bacteria, *Serratia marcescens*, and *Pseudomons tolaasii* were significantly suppressed by iHIL-E. In addition, although iHIL completely cleared *Salmonella* species at concentrations of >200 µg/100 µL, *Lactobacillus* species were unaffected by iHIL at concentrations of <1000 µg/100 µL. The present investigation shows that the devised automatic mass injection system is effective for the mass production of the extract of infected HIL and that this extract is a novel, natural, protease-resistant, antibiotic candidate with broad-spectrum antibiotic activity.

## 1. Introduction

Antibiotics play important roles in human and livestock health as they effectively treat and prevent infectious diseases caused by pathogens. Antibiotics have also been used to enhance animal growth and have enabled the development of intensive farming and the large-scale commercialization of livestock. On the other hand, the overuse and abuse of antibiotics in feed has caused many problems, such as the evolution of antibiotic-resistant bacteria, the presence of antibiotic residues in livestock products, weakening of disease resistance among livestock, and environmental pollution caused by antibiotic residues in excretions [1,2,3]. After recognizing the seriousness of the problems caused by antibiotic overuse and abuse, the European Union completely banned the use of in-feed antibiotics for growth promotion in 2006. The Korean government also prohibited in-feed antibiotics in July 2011. However, these governmental interventions are expected to increase therapeutic antibiotic use, because they increase the risks of livestock disease outbreaks and livestock death. Furthermore, as a result, chemical preservatives are being increasingly used as feed additives to prevent bacterial, fungal, and viral contamination. Hence, novel natural antibiotics are now being considered as a potential means of overcoming the problems associated with the overuse and abuse of antibiotics and chemical preservatives [4,5].

Insects can produce powerful antimicrobial peptides (AMPs) after activations of their innate immune systems by pathogen infection. Investigations suggest that insect AMPs are potent natural antibiotics that could be used to prevent and treat infectious diseases in man and livestock [6,7]. Pathogens activate cellular and humoral immunity systems in insects. Cellular immunity defends insects via the phagocytosis of pathogens such as bacteria, fungi, and protozoa, and by nodule formation and encapsulation. Humoral immunity protects insects with two different mechanisms the secretion of proteins or peptides targeting pathogens [8,9]. AMPs are pathogen-targeting peptides that are produced in fat and blood cells and secreted to hemolymph via the activation of humoral immunity in response to infections [7,10,11,12]. AMPs are categorized as cecropins, defensins, proline-rich peptides, lycine-rich peptides, and lysozymes, which are characterized according to their amino acid sequences and structures. To date, AMPs have been reported in a variety of insects orders such as Coleoptera, Diptera, Hymenoptera, and Lepidoptera [13,14,15,16,17,18].

We chose *Hermetia illucens* (the black soldier fly) as an AMP-producing candidate because it can synthesize powerful antimicrobials to protect it from pathogens in organic wastes such as food waste and livestock excrement [19] The larvae of *H. illucens* live in these organic wastes, which are heavily contaminated by bacteria, viruses, and fungi. Moreover, *H. illucens* has a relatively rapid life cycle as compared with other insects and feed costs are negligible. However, to trigger the mass production of AMPs, pathogens must be introduced into the body cavity because AMP expressions are up-regulated only after activation of the innate immune system [20,21]. In a previous study, we found that the antimicrobial activities of *H. illucens* larvae (HIL) extract and hemolymph were enhanced by infection with human or animal pathogens [22], and interestingly, Chol et al. reported that probiotics induce AMPs in HIL [23]. Here, we developed an automatic mass injection system (AMIS) to establish a means of producing an extract of infected *H. illucens* larvae (iHIL-E), and then evaluated the antimicrobial activities of iHIL-E against antibiotic- and AMP-resistant bacteria and pathogens, such as *Salmonella* species, and against *Lactobacillus* species.

## 2. Materials and Methods

### 2.1. H. illucens and Bacteria Strains

We obtained HIL from the Department of Agricultural Biology at the National Institute of Agricultural Sciences of the Rural Development Administration (Wanju, Korea) and maintained them under controlled conditions (room temperature (RT, 26 ± 1 °C) and 60% RH). We purchased *Escherichia coli* (KCCM 11234) and three *Lactobasillus* strains, that is, *Lactobacillus brevis* (KCCM 10553), *Lactobacillus casei* (KCCM 12413), and *Lactobacillus fermentum* (KCCM 11441) from the Korean Collection for Type Culture (Wonju, Korea). *Enterococcus faecalis* (KACC 11859), *Streptococcus mutans* (KACC16833), *Candida albicans* (KACC 30071), *Salmonella pullorum* (KVCC-BA0702509), *Salmonella typhimurium* (KCCM 40406), *Salmonella enteritidis* (KCCM 12021), *Serratia marcescens* (KACC11961), and *Pseudomonas tolaasii* (KACC15293) were bought from the National Institute of Animal Science of the Rural Development Administration to assess the antibiotic activity of HIL extracts (Wonju, Korea).

### 2.2. Induction of Antimicrobial Peptides (AMPs) in H. illucens Larvae (HIL) by Injecting L. casei

We injected *L. casei* into the abdomens of *H. illucens* fifth instar larvae using a fine needle (width 0.35 mm, length 40 mm) fixed to the AMIS (Figure 1A). Infected HIL were then starved for 24 h at room temperature. The gene expressions of cecropin 3 (HiCec3) and defensin 3 (HiDef3) in the hemolymph of infected HIL were analyzed by quantitative PCR. The Primer 3 program (http://simgene.com/Primer3) was used to design primers for HiCec3 and HiDef3. The *Drosophila melanogaster* actin 5C (DmAct5C) gene was used as an internal control. Primer sequences are presented in Table 1.

### 2.3. Preparation of H. illucens Larvae Extract

*L. casei*-infected HIL (4.5 kg) were dried by microwaving for 45 min and 1.5 kg of dried HIL was ground to a powder, which was suspended in 20% acetic acid solution, boiled for 30 min, and centrifuged at 4500 rpm for 30 min at 4 °C. The supernatant obtained was transferred to a new tube, dried for 9 h in a vacuum-spin drier, and dissolved in sterilized distilled-water at 50 mg/mL to produce infected *H. illucens* larvae extract (iHIL-E; final 306 g).

### 2.4. Radial Diffusion Assay

To evaluate iHIL-E protease resistance and evaluate the suitability of the AMIS for the mass production of *L. casei*-infected HIL, the antimicrobial activity of iHIL-E was assessed using a radial diffusion assay (RDA). First, bacteria were mixed with autoclaved liquid underlay gel (9 mM sodium phosphate, 1 mM sodium citrate, pH 7.4, 1% low electroendosmosis agar and 0.03% tryptic soy broth (TSB)) and the mixture was poured into a 100 mm square plate and allowed to harden. A 3.5 mm-diameter well was prepared on the underlay gel, 10 µL of iHIL-E incubated with or without 50 nM protease was loaded into the well, kept for 3 h at 37 °C, and then covered with sterilized overlay gel (6% TSB and 1% low electroendosmosis agar). The gel plate was then incubated for 18 h at 37 °C and then the diameter of the clear zone was measured and used to assess relative antimicrobial activity. Melittin (Sigma-Aldrich; Merck KGaA, Darmstadt, Germany) and purified bee venom (Chungjin Biotech Co., Ltd., Ansan, Korea) were used as controls.

### 2.5. Analysis of AMP Transcriptions

HIL were infected with *L. casei* and held at RT for defined times prior to total RNA extraction from hemolymph. To extract total RNA, HIL were transferred to sterilized microtubes and 1 mL of TRIzol (Thermo Fisher Scientific; Waltham, MA, USA) was added to each tube. HIL were then homogenized and total RNA was extracted according to manufacturer’s instruction. cDNA was synthesized using a high capacity reverse transcription kit (Applied Biosystems, Waltham, MA, USA), and the cDNA was used as a template to assess the gene expressions of cecropin 3 (HiCec3) and defensin 3 (HiDef3) in iHIL-E. We used the Primer 3 program (http://simgene.com/Primer3) to design primers for HiCec3 and HiDef3, and the *Drosophila melanogaster* actin 5C (DmAct5C) gene was used as an internal control to normalize gene expressions. Primer sequences are provided in Table 1.

### 2.6. Determination of the Minimum Inhibitory Concentrations (MICs) of HIL Extract versus Pathogens and Lactobacillus Species

Pathogens and *Lactobacillus* species were inoculated into liquid medium and cultured at 200 rpm in a shaking incubator at 37 °C to 4 × 10^6^ cfu/mL. Bacterial cultures were then diluted to 1 × 10^6^ cfu/mL placed in the wells of a 96-well plate and 90 µL/well of diluent was added. The bacterial cultures were then treated with 10 µL of serially diluted extracts and incubated for 18 h at 37 °C. Absorbances were measured at 600 nm to determine MICs. Absorbance in each well was normalized by the absorbance of diluted extract solution of the same concentration.

### 2.7. Statistical Analysis

Data from three independent experiments were analyzed using the Student’s *t*-test and one-way analysis of variance, and values are presented as means± standard deviations (SDs). Statistical significance was accepted for *p* values < 0.05.

## 3. Results

### 3.1. Analysis of Antimicrobial Activities and AMP Gene Expressions in iHIL-E

In a previous study, we reported that *L. casei* injection activated the immune system in HIL and induced AMP expression [23]. Therefore, we developed an AMIS using fine needles to inject HIL with *L. casei* in quantity (Figure 1A,D). The system was able to inject *L. casei* to 142500 HIL/h, but the manual method could inject it to 720 HIL/h (Figure 1C). The antimicrobial activities of iHIL-E using RDA and MIC were assessed. RDA showed that the antimicrobial activities of all HIL extracts were significantly enhanced by *L. casei* infection (Figure 1B). In contrast, when *L. casei* were injected by manual method into HIL, induced antimicrobial activities were observed in 86.1% of HIL extracts (Appendix A). We also evaluated whether the gene expressions of AMPs were induced by *L. casei* infection and found HiCec3 and HiDef3 gene expressions were dramatically enhanced by injection (Figure 2*)*. Furthermore, iHIL-E prevented the growths of *E. faecalis*, *S. mutans*, and *C. vaginitis* with MICs of 200, 500, and 1000 µg/100 µL, respectively (Figure 3).

### 3.2. Analysis of Protease Resistance of iHIL-E

The antimicrobial activities of AMPs can be substantially decreased by proteases [24]. Therefore, we assessed changes in the antimicrobial activity of iHIL-E caused by protease treatment using the RDA. The results showed that antimicrobial activities of melittin and purified bee venom were almost lost after treatment with 50 nM trypsin or 50 nM chymotrypsin for 60 min (Figure 4A). Furthermore, when 20 µg melittin or 20 µg purified bee venom was treated with trypsin plus chymotrypsin for 60 min or 24 h, their antimicrobial activities were completely lost (Figure 4A,B). In contrast, iHIL-E (5000 µg) was unaffected by trypsin or chymotrypsin after 60 min (Figure 4A). Moreover, iHIL activity was almost maintained after treatment with trypsin plus chymotrypsin for 1 h or 24 h (Figure 4A,B).

### 3.3. Analysis of Antimicrobial Activities of iHIL-E against Antibiotic- and AMP-Resistant Bacteria

The antimicrobial activities of iHIL-E against methicillin-resistant *Staphylococcus aureus* (MRSA) and multidrug-resistant *Pseudomonas aeruginosa* (MDRPA) was evaluated. The results showed that the growth of MRSA and MDRPA were dramatically inhibited by iHIL-E (MICs ~300 µg/100 µL; Figure 5A,B).

Some authors have reported the appearance of AMP-resistant bacteria [25,26,27], and thus, we evaluated the antimicrobial activities of iHIL-E against the AMP-resistant bacteria, *S. marcescens* and *P. tolaasii*. We found that the growths of both bacteria were effectively suppressed by iHIL-E (MICs ~300 µg/100 µL) but not by melittin (Figure 5C).

### 3.4. Confirmation of the L. casei Infection by AMIS

To confirm whether AMIS effectively infected HIL with *L. casei*, we checked the anti-microbial activity of iHIL-E after AMIS injection using *E. coli* (Gram-negative) and *S. aureus* and *Salmonella* species (*S. pullorum*, *S. tylpimurium* and *S. enteritidis*) (all Gram-positive) by determining MICs. We found that the growths of *E. coli* and *S. aureus* were completely suppressed by iHIL-E (MICs ~200 µg/100 µL; Figure 6A), and that it also inhibited the growth of *S. pullorum*, *S. tylpimurium*, and *S. enteritidis* (MICs ~200 µg/100 µL; Figure 6B). These results confirmed that AMIS induced antimicrobial activity in HIL by successfully introducing *L. casei* infection.

### 3.5. Analysis of the Antimicrobial Activity of iHIL-E against Lactobacillus Species

The antimicrobial activity of iHIL against the probiotics *L. brevis*, *L. casei*, and *L. fermentum*, was assessed to investigate its antimicrobial selectivity. Probiotics were completely destroyed by melittin at 1 µg/100 µL (Figure 7A). In contrast, the MIC of iHIL-E for *Lactobacillus* species was ~2000 µg/100 µL (Figure 7B). However, the viability of *Lactobacillus* species was significantly diminished by iHIL-E at 1000 µg/100 µL (Figure 7B). This result suggests that iHIL-E could be used to prevent contamination of fermented foods without seriously depleting probiotic levels.

## 4. Discussion

AMPs are well-known potent natural antibiotics and may in time replace synthetic antibiotics [28]. Furthermore, reports show that AMPs can be produced in abundance by insects [7,11,12,29] and that AMP production in insects is enhanced by bacterial infection [30,31,32]. In a previous study, we showed that the antimicrobial activities of and AMP expressions in hemolymph and in a HIL extract were significantly enhanced by bacterial infection [22,33]. These results suggest that insects could be used to produce AMPs in quantity and that iHIL-E has potential use as a commercial natural antibiotic.

Although natural antibiotics are attractive options in terms of overcoming antibiotics resistance and preventing environmental pollution, the industry requires that they can be produced cheaply and are biologically stable. Therefore, we developed an automatic mass injection system (AMIS) to reduce the cost of producing iHIL-E. Using this AMIS, 1000 HIL were perfectly injected at once and the iHIL-E extracts produced effectively suppressed the growths of pathogens like *E. coli*, *S. aureus*, and *S. tryphimurium* (Figure 1). As mentioned above, when *L. casei* was injected by manual method, antimicrobial activities were induced in 86.1% of HIL (Appendix A). The injection of *L. casei* into HIL by manual method is time-consuming and the induction rate of antimicrobial activity is also lower than that of AMIS. Furthermore, a lot of labor is necessary for mass production, and the amount of production per day is limited when the injection is performed by hand. In contrast, AMIS can inject *L. casei* into 2500 HIL at once without HIL death and failure of AMP induction. As shown in Figure 1C, to inject *L. casei* into HIL by the manual method, at least 19.79 persons are required to inject the same number of larvae as AMIS, if all persons can inject perfectly. Consequently, present investigation shows that AMIS is a novel and highly economical injection system for mass production of iHIL-E that can significantly reduce production cost by reducing production time and labor.

In this investigation, the result showed that the gene expressions of the antimicrobial peptides HiCec3 and HiDec3 were induced by *L. casei* infection in HIL (Figure 2). Moreover, iHIL-E exhibited strong antimicrobial activities against *E. faecalis*, *S. mutans*, *C. albicans*, *E. coli*, *S. aureus*, and *Salmonella* species (Figure 4 and Figure 6). Furthermore, iHIL-E completely blocked the growth of MRSA and MDRPA and of the antimicrobial peptide-resistant bacteria *S. marcescens* and *P. tolaasii* (Figure 5). These results indicate that the antimicrobial activity of iHIL-E was caused by the induction of AMP and that iHIL-E has a wide antimicrobial spectrum that extends from bacteria to fungi and can kill drug- and antimicrobial peptide-resistant bacteria.

The biological stabilities of AMPs are closely associated with their protease resistance. Many insect-originated AMPs involving melittin are easily degraded by proteases. Therefore, AMPs stable in the presence of proteases are effective natural antibiotic candidates because the shelf lives of foods are closely associated with the long-term stabilities of AMPs. In the present study, we found that the antimicrobial activity of iHIL-E was not significantly affected by trypsin and/or chymotrypsin treatment for 24 h, although both completely abrogated the antimicrobial activities of melittin and purified bee venom (Figure 3). In addition, our findings show that the antimicrobial activity of iHIL-E is caused by the inductions of AMPs (Figure 1 and Figure 2). Thus, our results show antimicrobial activity in iHIL-E is protease-stable and indicate that the extract has potential use as a protease-stable natural antibiotic. Moreover, our results demonstrate that AMPs in iHIL-E is resistant to protease. Several investigations showed that insect cecropins might be degraded by proteases secreted by pathogen, implying insect AMPs are sensitive to protease [34]. Evidences that endogenous protease inhibitors are present in insect hemolymph has been reported by several studies [35,36,37,38]. Therefore, although the results do not fully describe why the AMPs contained into iHIL-E are resistant to protease, it assumes that AMP itself is not resistant to protease, but AMPs in iHIL-E should be protected by an endogenous inhibitor in hemolymph from protease.

We also found that iHIL-E exhibited little toxicity against *Lactobacillus* species, such as *L. brevis*, *L. casei*, and *L. fermentum*, but potent activity against other bacteria and *C. albicans* (Figure 4, Figure 6, and Figure 7). In addition, it inactivates *E. faecalis*, *E. coli*, and *S. aureus*, and *Salmonella* species at 200 µg/100 µL and MRSA, MDRPA, *S. marcescens*, and *P. tolaasii* at 300 µg/100 µL (Figure 4A, Figure 5 and Figure 6). A narrow antimicrobial spectrum is a major disadvantage of natural preservatives, but iHIL-E completely inhibited the growths of various types of microorganisms, including bacterial and fungi, showed no antimicrobial activity against *Lactobacillus* species at concentrations <1000 µg/100 µL, and cleared antibiotic-resistant bacteria (Figure 5). These results demonstrate that iHIL-E may be used as a powerful potent natural preservative with little toxicity against probiotics.

## 5. Conclusions

In summary, this investigation showed that the iHIL-E prepared using the AMIS system exhibited potent wide spectrum, antimicrobial activities against bacteria and fungi, and was stable in the presence of proteases. Furthermore, the devised AMIS system enabled the production of iHIL-E in quantity. Taken together, this investigation shows that the AMIS system can substantially reduce the cost of producing iHIL-E and that this extract is a novel natural antibiotic with a wide range of applications, especially in the food, animal feed, and medicinal industries.

## Figures and Tables

**Figure 1 animals-10-02121-f001:**
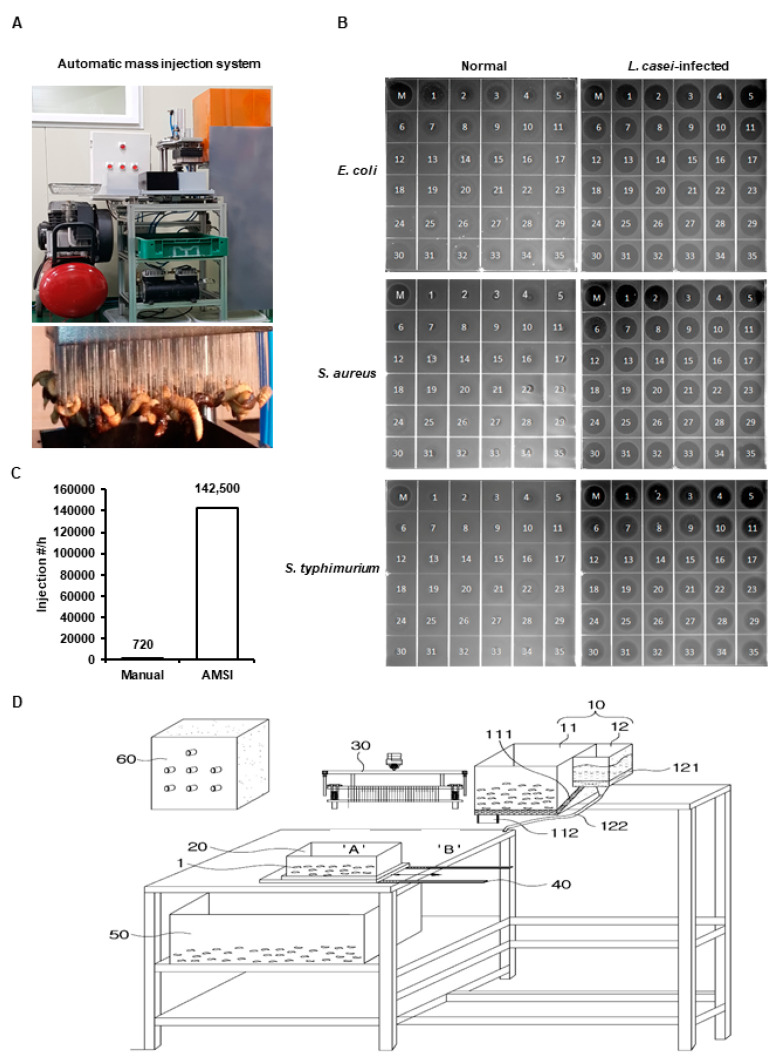
The automatic mass injection system (AMIS) and radial diffusion assay (RDA) results of the antimicrobial activities of iHIL-E against *E. coli*, *S. aureus*, and *S. typhimurium*. (**A**) *H. illucens* larvae (HIL) were injected by AMIS and held at RT for 24 h. (**B**) Antimicrobial activities of iHIL-E were determined by measuring the diameters of clear zones. (**C**) Comparison of injection numbers between the manual method and the AMIS method. Injection numbers of manual indicate injected-larvae number/h/person and those of AMIS indicate injected-larvae number/h/AMIS. (**D**) Schematic diagram of AMIS. (1) larvae (10) input, (11) larvae input, (12) inducer input, (20) loading part, (30) injector part, (40) transfer part, (50) collector, (60) main controller, (111) input controller, (112) discharge part, (121) bottom hopper, (122) discharge tube.

**Figure 2 animals-10-02121-f002:**
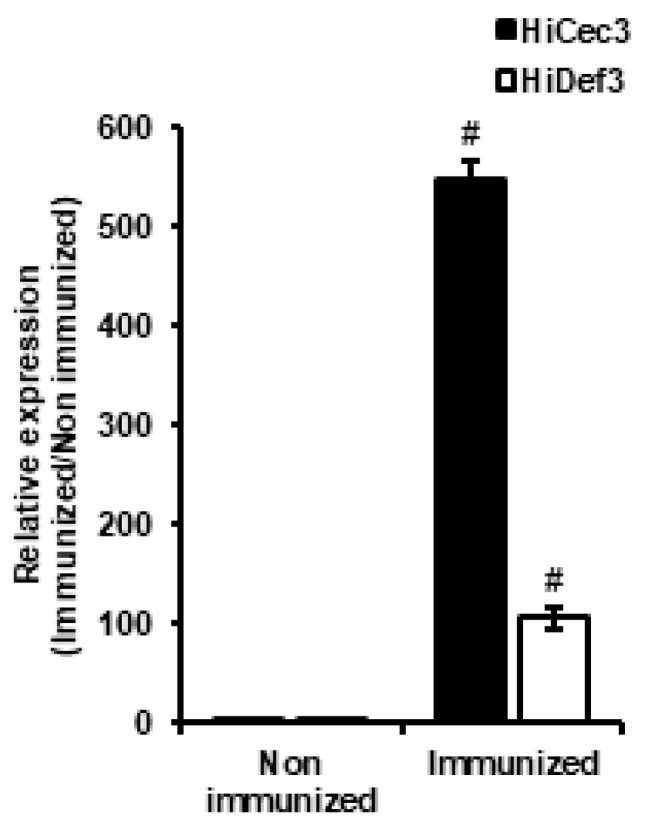
HiCeC3 and HiDef3 gene expressions in *L. casei*-infected HIL hemolymph. Gene expressions in hemolymph were evaluated 24 h after infection. *n* = 3. # defined *p* < 0.0001 vs. Non-immunized.

**Figure 3 animals-10-02121-f003:**
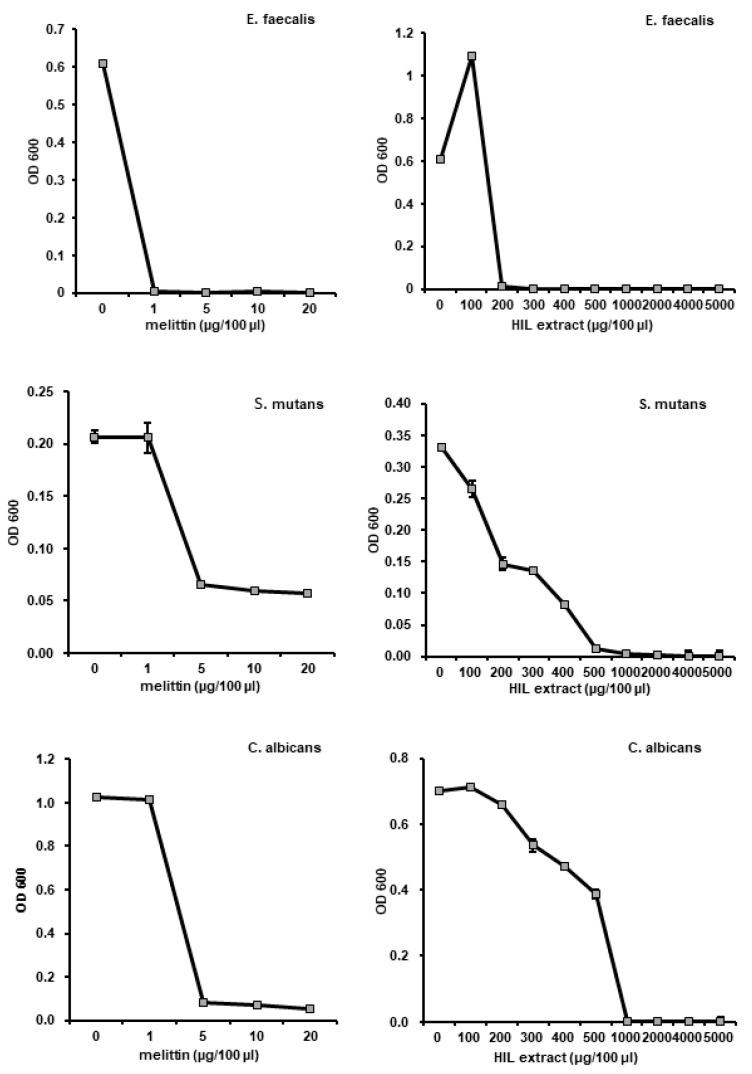
Evaluation of the antimicrobial activity of iHIL-E against *E. faecalis*, *S. mutans*, and *C. albicans*. Cells were treated with various concentrations of iHIL-E and incubated for 18 h at 37 °C in a shaking incubator. Cell viabilities are presented as optical densities at 600 nm.

**Figure 4 animals-10-02121-f004:**
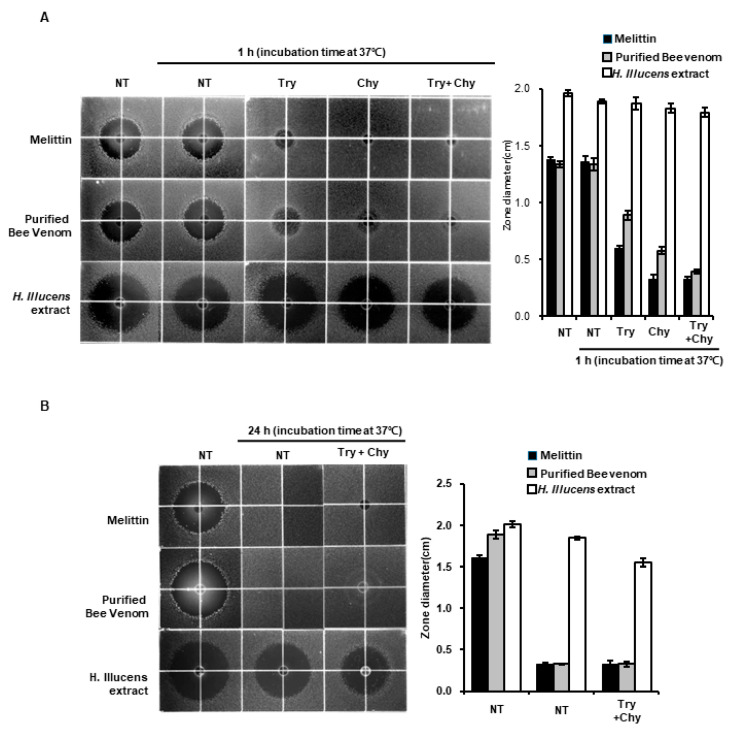
The effects of proteases on the antimicrobial activities of melittin (20 µg), purified bee venom (20 µg), and iHIL-E (5000 µg). Melittin, purified bee venom, and iHIL-E were treated with 50 nM trypsin, 50 nM chymotrypsin, or 50 nM trypsin & 50 nM chymotrypsin at 37 °C for (**A**) 1 or (**B**) 24 h. Antimicrobial activities were assessed using clear zone diameters.

**Figure 5 animals-10-02121-f005:**
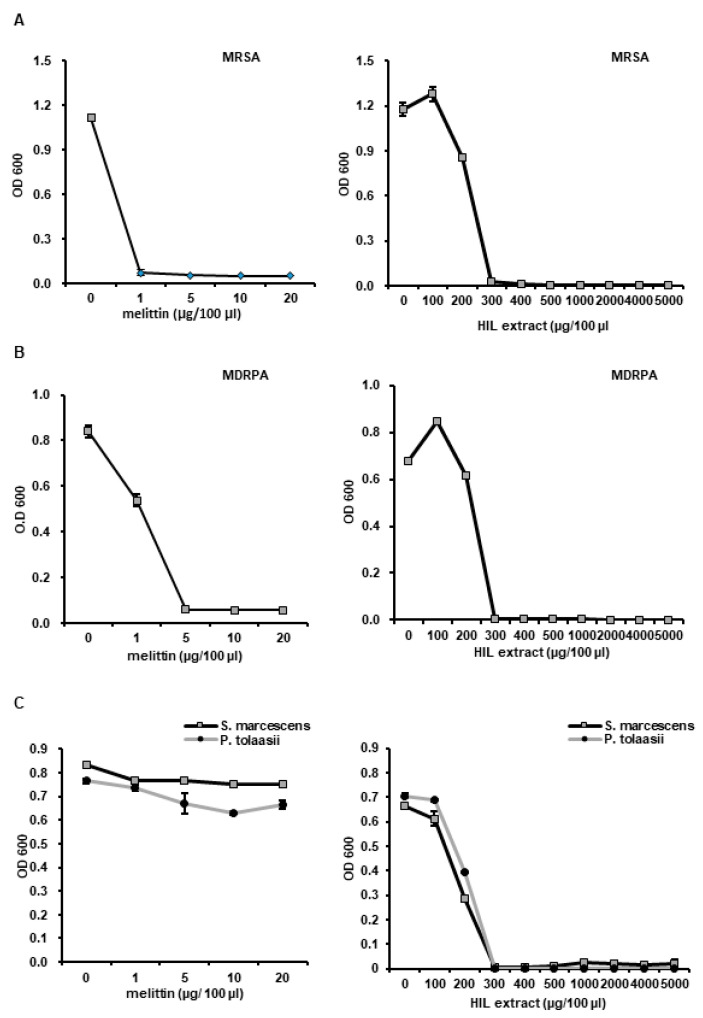
Evaluation of antimicrobial activities of iHIL-E against drug- and AMP-resistant bacteria. (**A**,**B**) MRSA and MDRPA were used as drug-resistant bacteria and (**C**) *S. marcescens* and *P. tolaasii* as AMP-resistant bacteria. (**A**–**C**) Bacterial cells were treated with iHIL-E at different concentrations for 18 h at 37°C in a shaking incubator. Cell viabilities are presented as optical densities at 600 nm.

**Figure 6 animals-10-02121-f006:**
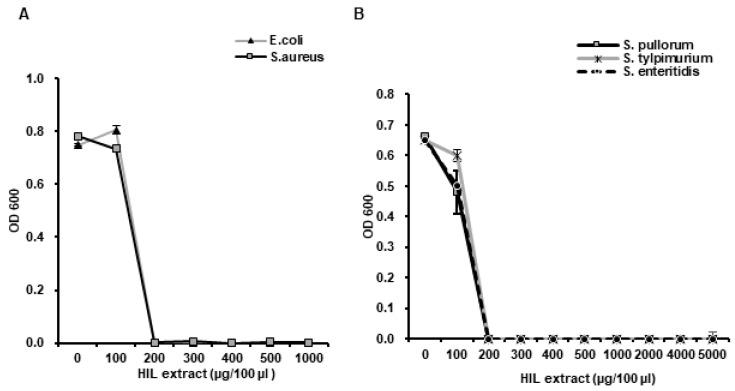
Evaluation of antimicrobial activities of iHIL-E produced using AMIS against *E. coli*, *S. aureus*, and *Salmonella* species. (**A**) Effects of iHIL-E on Gram-positive *S. aureus* and Gram-negative *E. coli*. (**B**) Effects of iHIL-E on *S. pullorum*, *S. tylpimurium*, and *S. enteritidis*. (**A**,**B**) Bacterial cells were treated with various concentrations of iHIL-E and incubated for 18 h at 37 °C in a shaking incubator. Cell viabilities are presented as optical densities at 600 nm.

**Figure 7 animals-10-02121-f007:**
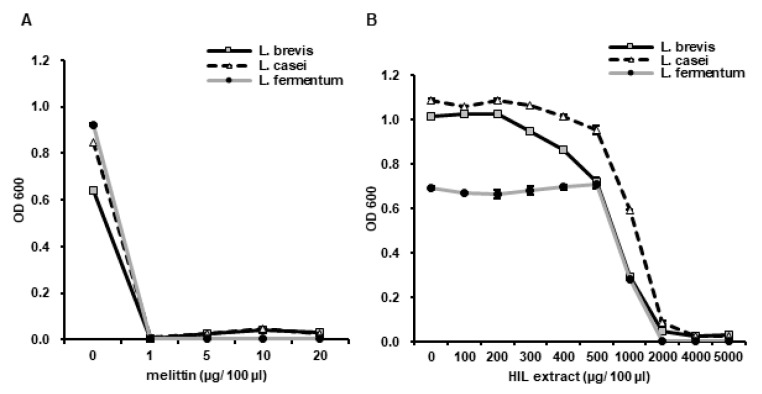
The selective antimicrobial activity of iHIL-E against *Lactobacillus* species. (**A**) Effects of melittin on *L. brevis, L. casei*, *and L. fermentum*. (**B**) Effects of iHIL-E on *L. brevis, L. casei*, *and L. fermentum*. (**A**,**B**) Bacterial cells were treated with various concentrations of melittin or iHIL-E for 18 h at 37 °C in a shaking incubator. Cell viabilities are presented as optical densities at 600 nm.

**Table 1 animals-10-02121-t001:** Primer sequences used for real-time polymerase chain reaction (PCR) of the HiCec 3 and HiDef 3 genes.

Name		Sequences
HiCec3	Forward	5′-ACCAGTGGAACGACTTGGTC-3′
	Reverse	5′-TCGAACCGTTGCCAGAACAT-3′
HiDef3	Forward	5′-GTGACCTCTTGAGCCCCTTC-3
	Reverse	5′-CTGCAGTTACAAACGGCTCG-3′
DmAct5C	Forward	5′-AAGGACTCGTACGTGGGTG-3′
	Reverse	5′-CATCTTCTCACGGTTGGC-3′

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
