# Peer review of "Evaluation of the Antimicrobial Activity of an Extract of Lactobacillus casei-Infected Hermetia illucens Larvae Produced Using an Automatic Injection System"

_animals, 2020, doi:10.3390/ani10112121_

Round 1

Reviewer 1 Report

After alterations, now manuscript can be published.

Author Response

Thank you for your indications.

We completely checked and corrected mistyped words in the manuscript. 

Reviewer 2 Report

Editor and Authors,

The revised version submitted by Lee et al. answered the critiques and suggestions; mainly, they provided better detailing of the biological material used and the methodology of the entire work. For example, Figure 1 is more informative.

There are species names not in italics: (L48, L. casei) in the main text and references (Apis mellifera, L330), (Bombyx mori, L332), and others. Also, journal titles should have the first letter in uppercase (For example, Annual review of entomology, L322) and others.

Sincerely,

Author Response

Thank you for your indications.

We completely corrected to italic or uppercases in the manuscript. 

Also, we corrected mistyped words in the manuscript.

Reviewer 3 Report

Dear authors,

Thanks for incorporating all the comments.

Kind regards,

One of the reviewers

Author Response

Thank you for your decision.

Now, we completely checked and corrected mistyped words in the manuscript. 

This manuscript is a resubmission of an earlier submission. The following is a list of the peer review reports and author responses from that submission.

Round 1

Reviewer 1 Report

The article entitle " Evaluation of the antimicrobial activity of an extract of Lactobacillus casei-infected Hermetia illucens 3 larvae produced using an automatic injection system " summarizes a broad investigation of antimicrobial activity of Lactobacillus and its production using an automatic system, a useful methodology with future application in antimicrobial production in large scale. The presented work, is focused and well supported by the obtained results.

The manuscript has been properly written and experiments and interpretations seem correct and legitimate. I therefore advice for publication after answer some questions. Below a point that merit some consideration:

In the beginning of the manuscript, it is explained that the aim of the work is to produce a Lactobacillus using an automatic injection system. Overall the authors applied a series of measurements complemented by their earlier results, the experiments are carefully described and the conclusion justified. Antibiotic properties were demonstrated but authors don’t compare the common methodology with this automatic injection system, statistics between both methodologies must be showed and compared, and must be emphasized in order to enhance the vantages of the proposed methodology, the aim of the work is not clear.

Author Response

Thanks for sharpen your indication and comment. Now, we reply for your comments as below.

The article entitle " Evaluation of the antimicrobial activity of an extract of Lactobacillus casei-infected Hermetia illucens 3 larvae produced using an automatic injection system " summarizes a broad investigation of antimicrobial activity of Lactobacillus and its production using an automatic system, a useful methodology with future application in antimicrobial production in large scale. The presented work, is focused and well supported by the obtained results.

The manuscript has been properly written and experiments and interpretations seem correct and legitimate. I therefore advice for publication after answer some questions. Below a point that merit some consideration:

In the beginning of the manuscript, it is explained that the aim of the work is to produce a Lactobacillus using an automatic injection system. Overall the authors applied a series of measurements complemented by their earlier results, the experiments are carefully described and the conclusion justified. Antibiotic properties were demonstrated but authors don’t compare the common methodology with this automatic injection system, statistics between both methodologies must be showed and compared, and must be emphasized in order to enhance the vantages of the proposed methodology, the aim of the work is not clear.

→ Thanks for your comment. We showed statistics between manual method and AMIS as Figure 1(C) and radial diffusion assay results of the antimicrobial activities of the extract of L.casei-injected HIL by manual method was obtained as Supplementary 1.

Reviewer 2 Report

The manuscript shows innavtive contents but the main problem is the discussion and the conclusions. Both are very poor. 

The simple summary must be rewritten is too short.

In the abstract, delete the first period (Previoulsy...): the abstract must be focused on the current research.

In the manuscript avoid the use of "we...."; use an impersonal form.

The discussion is very poor and, in general, is a repetition of the introduction, methods and results.

Author Response

Thanks for sharpen your indication and comment. Now, we reply for your comments as below.

The manuscript shows innavtive contents but the main problem is the discussion and the conclusions. Both are very poor. 

The simple summary must be rewritten is too short.

→ Thanks for your comments. We rewrote the simple summary in detail.

In the abstract, delete the first period (Previoulsy...): the abstract must be focused on the current research.

→ Thanks for your comments. We deleted the sentence.

In the manuscript avoid the use of "we...."; use an impersonal form.

→ Thanks for your comments. Most have been replaced to impersonal form.

The discussion is very poor and, in general, is a repetition of the introduction, methods and results.

→ Thanks for your sharpen indication. We discussed more detail and corrected and removed some sentences to prevent a repetition.

Reviewer 3 Report

Dear Editor and Authors,

Lee et al. describe an experiment using an automated system to inject microorganisms into insect larvae to stimulate antimicrobial peptides production. The production and action of these peptides against pathogens are investigated through several experiments. Also, the non-action of these AMP on Lactobacillus was verified. The experiments are robust, and the results very interesting. However, the automated system's functioning is not sufficiently described, except for the photo in Figure 1. Suppose this is an essential finding of the research, it should be better described.

Below are listed some points that should be corrected before publication.

L35 – H. illucens – please, change to italic;

L66 – "variety of insect orders";

L68 – "as an AMP-producing";

L68-70 - The reason for choosing this species is interesting. Please provide references that support the hypothesis that insects that feed on waste produce more potent antimicrobials;

L83 - As a matter of style, most journals advise authors not to start any sentence with an abbreviation or a number, including species name;

L86 - Please provide the identification number of these isolates;

L92 and L99 - In these cases, species names must be different from the rest of the text;

L99 - Please add the initial quantities (number of larvae, mass) that were used in order to obtain the yield calculation;

L100 – idem to comment L83;

L132-136 - Did the extracts have any degree of absorbance that could affect the reading of the microplates? In this case, more concentrate extracts could affect absorbance reading.

Author Response

Thanks for sharpen your indication and comment. Now, we reply for your comments as below.

Lee et al. describe an experiment using an automated system to inject microorganisms into insect larvae to stimulate antimicrobial peptides production. The production and action of these peptides against pathogens are investigated through several experiments. Also, the non-action of these AMP on Lactobacillus was verified. The experiments are robust, and the results very interesting. However, the automated system's functioning is not sufficiently described, except for the photo in Figure 1. Suppose this is an essential finding of the research, it should be better described.

Below are listed some points that should be corrected before publication.

L35 – H. illucens – please, change to italic;

→ Thanks for your comment. We corrected.

L66 – "variety of insect orders";

→ Thanks for your comment. We corrected.

L68 – "as an AMP-producing";

→ Thanks for your comment. We corrected.

L68-70 - The reason for choosing this species is interesting. Please provide references that support the hypothesis that insects that feed on waste produce more potent antimicrobials;

→ Thanks for your comment. We provided the reference.

L83 - As a matter of style, most journals advise authors not to start any sentence with an abbreviation or a number, including species name;

→ Thanks for your comment. We corrected the sentence

L86 - Please provide the identification number of these isolates;

→ Thanks for your comment. We mentioned the identification number of all bacteria.

L92 and L99 - In these cases, species names must be different from the rest of the text;

→ Thanks for your comment. We checked species names and corrected completely.

L99 - Please add the initial quantities (number of larvae, mass) that were used in order to obtain the yield calculation;

 → Thanks for your comment. We added the initial quantity as mass and obtained the mass of final extract in the sentences.

L100 – idem to comment L83;

→ Thanks for your comment. We corrected the sentence

L132-136 - Did the extracts have any degree of absorbance that could affect the reading of the microplates? In this case, more concentrate extracts could affect absorbance reading.

→ Thanks for your comment. We mentioned how to be normalized the absorbance of each sample.

Reviewer 4 Report

Dear authors,

I have indicated my comments in attached file. Main comments:

  • Recommend to provide schematic diagram of AMIS
  • Provide numbers to justify the cost effectiveness and speed of AMIS
  • AMPs developed during this study are protease resistant. No justification provided regarding why they are resistant?

Thanks and regards,

One of the reviewer

Author Response

Thanks for sharpen your indication and comment. Now, we reply for your comments as below and as attached file.

I have indicated my comments in attached file. Main comments:

  • Recommend to provide schematic diagram of AMIS

→Thanks for your comment. We provided schematic diagram of AMIS as Figure 1(D).

  • Provide numbers to justify the cost effectiveness and speed of AMIS

→ Thanks for your comment. We showed injection numbers/h of manual method and AMIS, respectively, as Figure 1(C). And we also showed RDA results of the antimicrobial activities of the extract of L.casei-injected HIL by manual method to justify the effectiveness of AMIS.

  • AMPs developed during this study are protease resistant. No justification provided regarding why they are resistant?

→ Thanks for your comment. We discussed why AMPs in iHIL-E is resistant to protease in discussion section and provided references.
